# Associations among Serum Lipocalin-2 Concentration, Human Papilloma Virus, and Clinical Stage of Cervical Cancer

**DOI:** 10.3390/medicina55060229

**Published:** 2019-05-30

**Authors:** Agnė Vitkauskaitė, Joana Celiešiūtė, Saulius Paškauskas, Erika Skrodenienė, Rūta Jolanta Nadišauskienė, Aušra Burkauskienė, Daiva Vaitkienė

**Affiliations:** 1Department of Obstetrics and Gynaecology, Medical Academy, Lithuanian University of Health Sciences; LT 44307 Kaunas, Lithuania; joana.celiesiute@gmail.com (J.C.); sauliuspaskauskas@yahoo.com (S.P.); ruta.nadisauskiene@gmail.com (R.J.N.); vaitkiene.daiva@gmail.com (D.V.); 2Department of Laboratory Medicine, Medical Academy, Lithuanian University of Health Sciences, LT 44307 Kaunas, Lithuania; erika.skrodeniene@kaunoklinikos.lt; 3Institute of Anatomy, Medical Academy, Lithuanian University of Health Sciences, LT 44307 Kaunas, Lithuania; ausra.burkauskiene@gmail.com

**Keywords:** cervical cancer, human papillomavirus, lipocalin 2

## Abstract

*Background and objective:* Lipocalin 2 (LCN2) has an oncogenic role in promoting tumorigenesis through enhancing tumor cell proliferation and the metastatic potential. The aim of our study was to determine whether serum LCN2 could serve as a diagnostic marker of cervical cancer (CC) and to evaluate the correlation between its serum concentration, the clinical stage of the cancer and Human Papilloma Virus HPV infections in women. *Materials and methods:* A total of 33 women with histologically proven cervical cancer (CC), 9 women with high- grade cervical intraepithelial neoplasia (HSIL) and 48 healthy women (NILM) were involved in the study. A concentration of LCN2 was assayed with the Magnetic LuminexR Assay multiplex kit. An HPV genotyping kit was used for the detection and differentiation of 15 high-risk (HR) HPV types in the liquid-based cytology medium (LBCM) and the tissue biopsy. *Results:* The majority (84.8%) of the women were infected by HPV16 in the CC group, and there was no woman with HPV16 in the control group (*P* < 0.01). Several types of HR HPV were found more often in the LBCM compared to in the tissue biopsy (*P* = 0.044). HPV16 was more frequently detected in the tissue biopsy than the LBCM (*P* < 0.05). The LCN2 level was higher in HPV-positive than in HPV-negative women (*P* = 0.029). The LCN2 concentration was significantly higher in women with stage IV than those with stage I CC (*P* = 0.021). Conclusions: Many HR HPV types, together with HPV16/18, can colonize the vagina and cervix, but often HPV16 alone penetrates into the tissue and causes CC. The serum LCN2 concentration was found to be associated not only with HR HPV infection, irrespective of the degree of cervical intraepithelial changes, but also with advanced clinical CC stage. LCN2 could be used to identify patients with advanced disease, who require a more aggressive treatment.

## 1. Introduction

The majority of cervical cancer (CC) cases occur as a result of the effect of human papillomavirus (HPV) infection. However, only a small proportion of women with HPV infection and intraepithelial lesions progress to CC development. It is therefore believed that there are different risk factors contributing to the development and progression of this cancer. HPV infection alone is not enough for the complicated processes of cellular change. Tumor formation is a multistage process, the development of which requires other important factors such as genetic damage or mutations, as well as environmental factors [1]. One of the stages of cancer development is inflammation and the body’s immune response to the inflammation [2]. Inflammation in the pathogenesis of cancer development has a dual role: although it is necessary for the onset of anti-cancer reactions, at the same tame it can promote cancer progression and growth. Genetic differences associated with an individual’s immune system may be important in responding to HPV and may increase or decrease the risk of CC development [3].

Lipocalins (LCNs) are proteins that perform different functions in the regulation of the cell metabolism, immune response, prostaglandin synthesis and others. LCN2 is a glycoprotein released by various cells of the organism, including epithelial cells, macrophages, neutrophils and tumor cells, and this protein can be found in plasma, serum, and urine in different clinical situations, including metastasis of breast cancer or colon cancer [4]. Greater concentrations of LCN2 can be detected in patients with breast metastasis, which shows that it can be viewed as a cancer marker and that it relates to cell proliferation [5].

LCN2, as an acute phase protein, may also be involved in a variety of physiological processes, such as iron transport, prostaglandin synthesis, and the secretion of inflammatory markers, which have been identified in the study on mice and can improve cell survival functions [6]. LCN2 may indirectly be associated with uterine cancer, mediated by IL-8 secretion. Lin et al. concluded that the overexpression of LCN2 in the uterus might be related to the formation of uterine cancer, because it increases the production of inflammatory markers. Obviously, more studies are needed to further elucidate the possible roles of LCN2 in tumorigenesis in female diseases [6].

Kim et al. showed a new role for LCN2 as a glycolytic switch in colorectal cancer cells. The results of their study revealed the role of LCN2 major molecular mechanisms in the progression of colorectal cancer, indicating LCN2 as a possible diagnostic marker and a therapeutic target in colorectal cancer [7]. The findings by Mannelqvist et al. identified relationships between the LCN2 expression and aggressive cancer properties, long-term metastatic spread and decreased patient survival in this cancer [8]. The findings by Syrjänen et al. showed that LCN2 expression increased with the degree of squamous intraepithelial lesions. However, the expression pattern and role of LCN2 in CC are still poorly understood [9].

The aim of our study was to determine the serum concentration of LCN2 in CC patients and healthy women with different HPV types as a possible diagnostic marker and to evaluate associations between its serum concentration and the clinical stage of CC.

## 2. Materials and Methods

### 2.1. Patients

A total of 33 women with histologically proven squamous cells CC and 9 women with high-grade cervical intraepithelial neoplasia, treated in the Department of Obstetrics and Gynaecology, Hospital of Lithuanian University of Health Sciences during 2017–2018, were involved in this study. A revised FIGO staging classification of carcinoma of the cervix uteri was used (2018).

The control group consisted of 48 healthy women without cervical intraepithelial lesions or malignancy confirmed by the liquid-based cytology test (SurePath, Becton Dickinson, Franklin Lakes, NJ, USA). The results of the cervicovaginal cytology test were reported by the 2014 Bethesda System: negative for intraepithelial lesions or malignancy (NILM) or for lesions and high-grade squamous intraepithelial lesions (HSIL) [10].

Patients were excluded from the study if they were aged ≤ 18 years, were pregnant, had any autoimmune disease, diabetes and cancer other than CC, or had previously received treatment for carcinoma in situ or HSIL. All patients signed their informed consent before being included in the study. The study was approved by the Kaunas Regional Biomedical Research Ethics Committee (No. BE-2-80/2018).

### 2.2. Blood Sampling and Determination of Serum Concentration of Inflammatory Markers 

The patients were asked to provide a blood sample before the surgery and cancer treatment. The blood specimens were coded to maintain confidentiality, were frozen and were stored at −80 °C until use. The study population blood was analyzed for LCN2. The concentration of LCN2 was assayed with the Magnetic LuminexR Assay multiplex kit (Human Premixed Multi-Analyte Kit, R&D Systems, Inc., Minneapolis, MN, USA). Magnetic microparticles were coated with test antibodies. The specimens, standards, and microparticles were pipetted into test wells and joined existing antibodies. An antibody mixture specific to the tested markers was added to each well. After washing, a streptavidin-phycoerythrin conjugate was added to each well, and the microparticles were resuspended in buffer and read using the Luminex 100 system (Luminex, Austin, TX, USA). A sample from each well was imaged with a CCD camera, with a set of the filter to differentiate the excitation levels.

### 2.3. HPV DNA Detection and Typing

The HPV DNA detection was performed using a QIAamp DNA mini kit (QIAGEN, Germantown, MD, USA). An HPV genotyping (DiaMex, Miami, FL, USA) kit was used for the qualitative and quantitative detection and differentiation of 15 high-risk (HR) HPV types and 3 putative HR types in the liquid-based cytology medium SurePath (Becton Dickinson, Franklin Lakes, NJ, USA) using the real-time hybridization-fluorescence detection of amplified products (Luminex Corporation, USA). The hybridization method is based on the simultaneous real-time amplification of DNA fragments of the HPV genotypes 16, 18, 31, 33, 35, 39, 45, 51, 52, 56, 58, 59, 68, 73, 82 and 26, 53, 66, as well as a DNA fragment of the β-globin gene (used as an internal endogenous control) in one tube. Each genotype was detected in a separate fluorescent channel. A quantitative analysis was performed in the presence of DNA calibrators.

### 2.4. Statistical Analysis

A statistical data analysis was performed using the statistical package IBM SPSS 23.0. The Kolmogorov-Smirnov test was employed to determine if the quantitative data were normally distributed. All data were found to be non-normally distributed. The non-normally distributed data were compared using the Mann-Whitney *U* test and Kruskal-Wallis test. The chi-square test and the Fisher exact test (for a small size sample) were used to determine whether a relationship exists between the qualitative data. The proportions were compared using the *z* test. Differences in comparing the groups were considered statistically significant when a *P* value was under 0.05. The study power analysis was done and determined as 0.8.

## 3. Results

A total of 90 women aged 25–85 years were included in the analysis. The median age of the women, by different intraepithelial findings and CC stages, is shown in Table 1. All women in the CC and HSIL groups were HPV positive, and this percentage significantly differed from the percentage of HPV-positive women in the NILM group (100% and 100% vs. 20.8%, *P* < 0.001) (Table 2). 

We classified HPV16 and HPV18 together (HPV16/18), because they are the types that most often cause cervical lesions; other high-risk HPV types, which were most commonly found in our study population; and HPV16/18 together with other types. The overall prevalence of HPV according to cervical intraepithelial abnormalities is shown in Table 3.

The statistical analysis was performed by a chi-square test and Fisher exact test when the sample sizes were small. The proportions are compared using a *z* test.

The prevalence of individual HPV types among CC and healthy women is presented in Figure 1. The majority of women in the CC group were infected with the HPV16 type alone or in combination with other HR HPV types. In the CC group, 84.8% of the women were infected by HPV16, and none of the women were found to have this HPV type in the control group (*P *< 0.001). The other HPV types that were most frequently detected in CC and healthy women were the types HPV18, HPV31, and HPV52. HPV39, HPV51, and HPV58 were the most common types among the control women (20% of each type, Figure 1).

The statistical analysis was performed by a chi-square test and Fisher exact test when the sample sizes were small. The proportions are compared using a *z* test.

We compared the prevalence of HPV types in the liquid-based cytology medium (LBCM) and in the tissue in the same CC women group. One HR HPV type was detected in 60.6% (n = 20), two in 30.3% (n = 10), and three in 9.1% (n = 3) of the cases in LBCM. We detected a lower number of other HPV types in the tissue samples compared to LBCM (Figure 2). In the tissue biopsy, HR HPV of one type was documented more frequently when compared to two types and more (90.6%, n = 29 and 9.4%, n = 3, respectively; *P* = 0.015). More than one HR HPV type was found more frequently in LBCM when compared to the tissue biopsy (39.4%, n = 13 and 9.4%, n = 3, respectively; *P* = 0.044) (Figure 3). HPV16 alone and/or HPV18 types were detected more frequently in the tissue biopsy than the LBCM (84.8%, n = 28 vs. 57.6%, n = 19; *P* < 0.05). 

The statistical analysis was performed by a chi-square test and Fisher exact test when the sample sizes were small.

The statistical analysis was performed by a chi-square test and Fisher exact test when the sample sizes were small. 

We analyzed the LCN2 concentration in HSIL, CC cases, HPV-positive and HPV-negative controls and found that the LCN2 level was significantly higher in HPV-positive than in HPV-negative women (median 32,983 (25th–75th percentile, 10,008–41,140) pg/mL and 15,540 (25th–75th percentile, 1237–32,872) pg/mL, respectively; *P* = 0.029), but no difference was observed between HPV16/18 and other HPV types (Figure 4).

The statistical analysis was performed by a Mann-Whitney *U* test. The horizontal dashes represent the medians.

A comparison of the circulating concentrations of LCN2 according to the HPV status and the degree of cervical intraepithelial abnormalities (Figure 5) showed that the LCN2 level was higher in healthy HPV-positive than in healthy HPV-negative women, but the difference did not reach a statistical significance (median 39,374 (25th–75th percentile, 7778–41,147) and 15,540 (25th–75th percentile, 1237–32,872) pg/mL, respectively; *P* = 0.087). No significant difference was found in the LCN2 level when comparing the HSIL and CC patients to the healthy HPV-negative women as well. The median LCN2 concentration in the HSIL group was 37,660 (25th–75th percentile, 21,698–41,385) pg/mL, and in the CC group it was 32,010 (25th–75th percentile, 1271–41,665) pg/mL.

The statistical analysis was performed by a Mann-Whitney *U* test. The horizontal dashes represent the medians.

The LCN2 concentration in the CC patients was significantly higher in women with stage IV compared to stage I (40,036 (25th–75th percentile, 32,810–49,751) and 1044 (25th–75th percentile, 1042–) pg/mL, respectively; *P *= 0.021), but did not reach a statistical significance between the other CC stages. The median LCN2 concentration was 29,950 (25th–75th percentile, 1262–41,666) pg/mL in women with II stage CC and 20,378 (25th–75th percentile, 10,071–40,820) pg/mL in those with III stage CC (Figure 6).

The statistical analysis was performed by a Mann-Whitney *U* test. The horizontal dashes represent the medians.

## 4. Discussion

In this study, HPV16 was found to be the most prevalent HPV type in the CC women. A study by Félix et al., retrospectively analyzing 1214 tumor samples from 1928 to 2005, reported that the most common HPV16 and HPV18 accounted for almost 70% of CC in the Portuguese female population in all 9 decades that were studied [11]. HR HPV is an important risk factor in the progression of cervical intraepithelial lesions and in the development of CC. The identification of HPV characteristics and types is crucial for the implementation of CC prevention and control strategies [12]. However, the results from different studies show that there is not always a clear association between HPV infection and intraepithelial lesions. Although HPV is recognized as an important risk factor in the progression of intraepithelial lesions and the development of CC, accumulating evidence suggests that HPV is probably not the main reason for the development of carcinoma in situ [13,14,15].

We analyzed the systemic level of LCN2 associated with chronic cervical inflammation during HPV infection as a potential biomarker to identify patients with a different CC stage. Studies have shown that LCN2 plays an important role in the pathogenesis and metastasis of various types of cancer [6,16]. The LCN2 serum concentration is easily analyzed and can be used for an early diagnosis of CC [17]. When CC cases were classified according to the FIGO stages, the LCN2 levels were significantly higher in patients with stage IV CC than in those with early stage I CC. However, the LCN2 concentrations between patients with early stage (I/II) CC and those with healthy controls did not differ [17].

Although LCN2 has an oncogenic role in promoting tumorigenesis, it is not a specific marker. High concentrations of LCN2 in serum can be found in various diseases, especially in the case of inflammation developing, infection, or tissue ischemia. LCN2 can enhance the local protective immune barrier against various pathogens [18]. This protein can be detected in patients with obesity, because it is associated with metabolic syndrome and inflammation and plays as a regulator of multiple responses at different levels. High concentrations of LCN2 are found in the serum of patients with rheumatic and kidney diseases [19]. 

In their study, Sánchez-Zauco et al. evaluated circulating cytokines and chemokines as potential diagnostic biomarkers for gastric cancer (GC). The study included healthy blood donors and found that IL-6, IFN-γ, and IL-10 might be associated with GC and can be used as markers to identify patients at potential risk for GC [20]. In our study, we investigated healthy women without cervical intraepithelial abnormalities (NILM), which indicated that they do not have inflammation, infection, or another metabolic disease at the moment of the blood collection, but no objective investigation has been done to reject co-morbidities. 

The role of LCN2 in HR HPV-associated CC was analyzed in a study by Syrjänen et al. [9]. The findings of this study showed that LCN2 was related to the outcome of HPV. LCN2 was found to be an important marker that helped to detect the incidence of HPV infection, but it was not sensitive enough to detect HPV persistence or disappearance [9]. Our study also determined that the LCN2 concentration was higher in HPV-positive women, independent of the degree of cervical intraepithelial abnormalities or the HPV type.

The upregulation of LCN2 in intraepithelial lesions is induced by the E6 protein of HR HPV [17]. In the study by Syrjänen et al., LCN2 up-regulation was most consistently associated with high-grade intraepithelial lesions [9]. The amount of this lipocalin was also closely related to the determination of HPV and the virus content. LCN2 expression had no associations with HPV outcomes, but LCN2 could be considered as one of the new cell proteins regulated by oncogenic HPV. 

The study by Chung et al. showed that the increased levels of LCN2 were associated with lymph node involvement in patients with CC. LCN2 can improve invasion abilities in CC cell lines and can activate cancer cell metastasis by increasing cancer cell motility through the activation of mesenchymal transition pathway components [21]. The role of LCN2 in promoting the tumorigenesis of other cancers is also well known. An increased concentration of LCN2 in cholangiocarcinoma cells was associated with a higher frequency of metastasis in such patients’ groups and a worse prognosis. Chiang et al. suggested LCN2 as a real target for cholangiocarcinoma eradication [22]. 

LCN2 can activate the malignant properties of breast cancer cells. The study by Ören et al. reported the tumorigenic potential of LCN2 and provided evidence that LCN2 could be a promising therapeutic goal [23]. In their study, Manelqist et al. formulated a hypothesis that breast cancer progression may be based on the presence of LCN2 in the tumor-supportive microenvironment [8].

Our study data support other authors’ findings that LCN plays an important role in the oncogenesis of cancer, but it is not an early and specific marker of CC. Because of its pro-proliferative, pro-regenerative, and anti-inflammatory properties, LCN2 might be of great importance in the progression of cervical and other cancers, but the diverse biological effects of LCN2 within the tumor-supportive microenvironment, along with other factors besides HR HPV that can affect lipocalin levels, should be further studied.

## 5. Conclusions

The present study shows that many HR HPV types, together with HPV16/18, can colonize the vagina and the cervix, but that HPV16 often penetrates in the tissue alone and causes CC. The presence of HR HPV significantly increased the LCN2 level, independently of the degree of cervical intraepithelial changes. The significantly elevated LCN2 serum concentration was associated with an advanced clinical CC stage but did not differ between women with early stages (stages I–II) and healthy controls. LCN2 could be used to identify patients with an advanced disease who require a more aggressive treatment. 

## Figures and Tables

**Figure 1 medicina-55-00229-f001:**
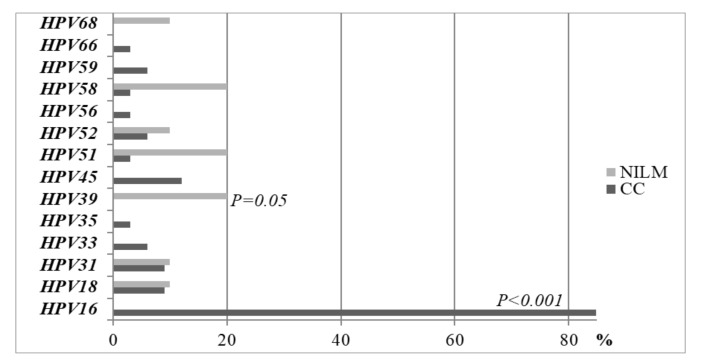
Prevalence of HPV types between CC women and healthy controls in the LBC medium.

**Figure 2 medicina-55-00229-f002:**
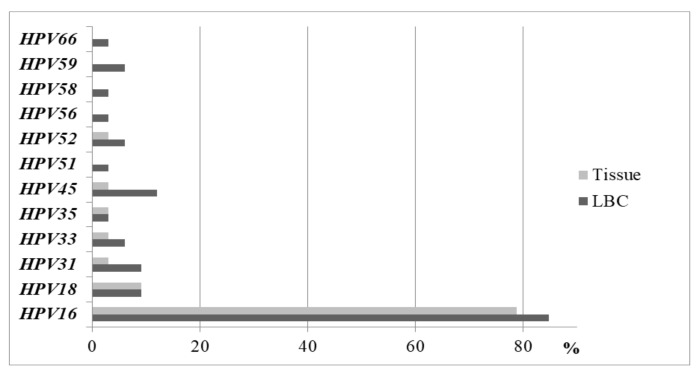
Prevalence of HPV types in the LBC medium and in the tissue in the CC women group.

**Figure 3 medicina-55-00229-f003:**
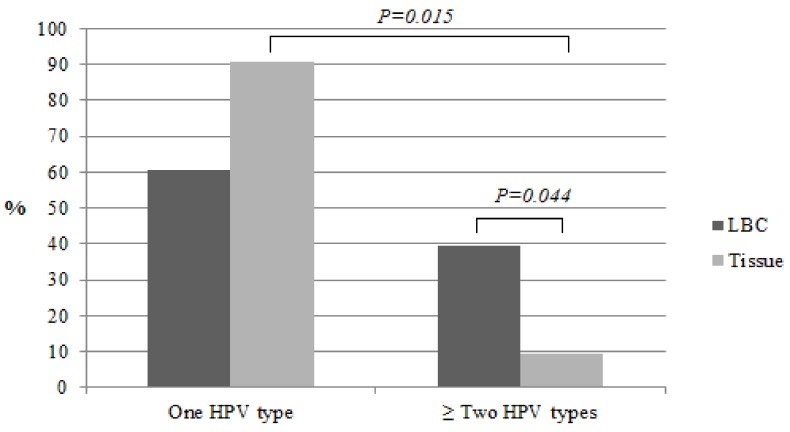
Prevalence of HPV types in LBC and in the cervical tissue in CC women.

**Figure 4 medicina-55-00229-f004:**
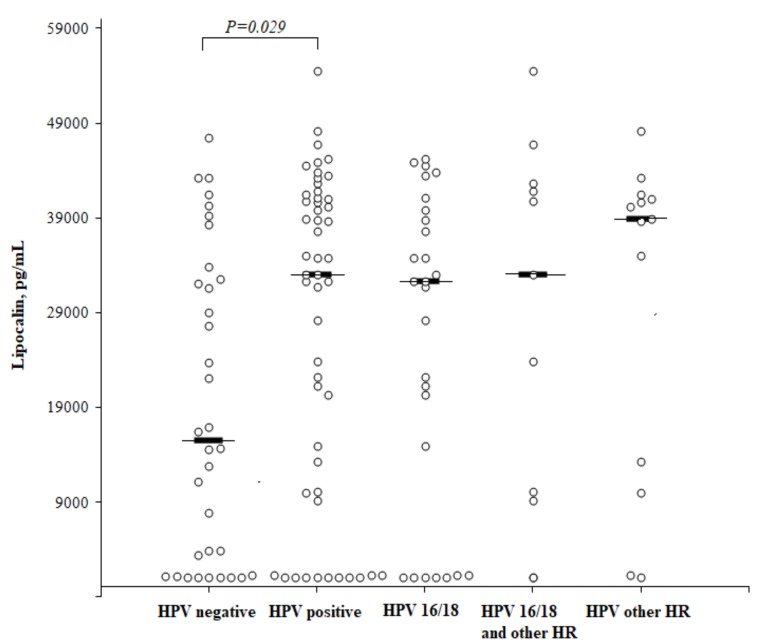
The circulating concentrations of lipocalin according to the HPV status. High-risk (HR).

**Figure 5 medicina-55-00229-f005:**
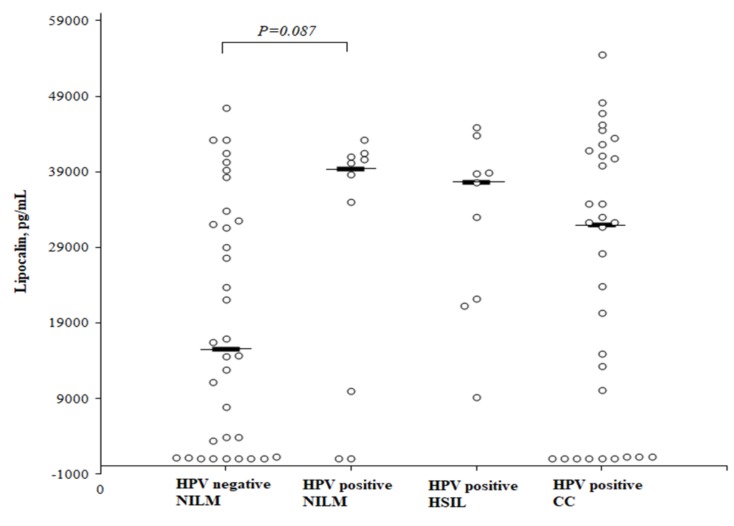
The circulating concentrations of lipocalin according to the cervical intraepithelial abnormalities and HPV status.

**Figure 6 medicina-55-00229-f006:**
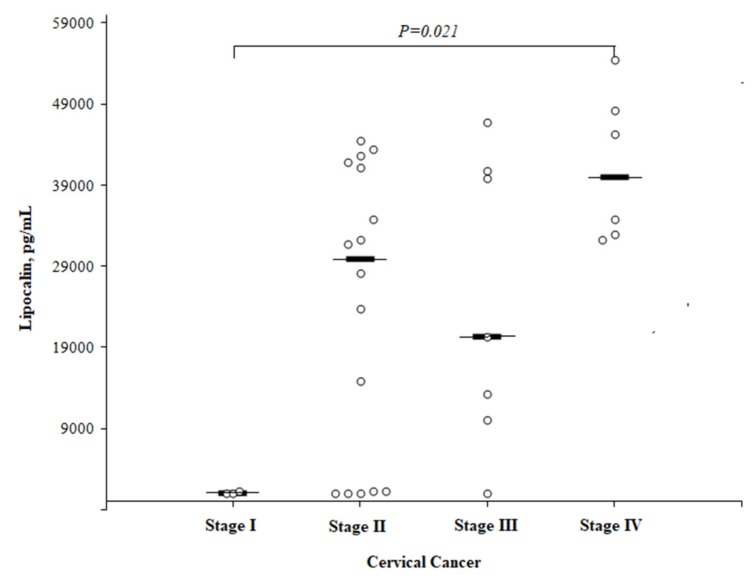
The circulating concentrations of lipocalin by different cervical cancer stages.

**Table 1 medicina-55-00229-t001:** The age of women by different intraepithelial findings and cervical cancer CC stage.

Population	n	Age, Median (25th–75th Percentile),Years	Age Range, Years	*P*
NILM	48	48 (44–54)	25–78	0.041
HSIL	9	45 (33–53)	29–60	
CC	33	53 (44–66)	30–85	
Stage I CC	3	43 (30-)	30–45	0.072
Stage II CC	17	59 (46.5–69.5)	32–74	
Stage III CC	7	48 (43–58)	41–62	
Stage IV CC	6	64 (47–78.25)	38–85	
Total	90	50 (43–59)	25–85	

The statistical analysis was performed by a Kruskal-Wallis test. High- grade cervical intraepithelial neoplasia (HSIL); negative for intraepithelial lesions or malignancy (NILM).

**Table 2 medicina-55-00229-t002:** Characteristics of the cervical cancer patients and healthy women, and the prevalence of HPV in the liquid-based cytology medium (LBC) medium.

Population	HPV Negative n (%)	HPV Positive n (%)
NILM (N = 48)	38 (79.2) ^a^	10 (20.8) ^b^
HSIL (N = 9)	0 ^c^	9 (100) ^d^
CC (N = 33)	0^c^	33 (100) ^d^

*P *< 0.001, ^a^ compared with ^b^; ^a^ compared with ^c^; ^b^ compared with ^d^, and ^c^ compared with ^d^. Human Papilloma Virus HPV

**Table 3 medicina-55-00229-t003:** Characteristics of the cervical cancer patients and healthy women included in the study, and the prevalence of HPV types in the LBC medium.

Population	HPV16/18 n (%)	HPV16/18 and Other Types n (%)	Other HPV Types n (%)
NILM (N=48)	1 (2.1) ^a^	0 ^c^	9 (18.8)
HSIL (N=9)	7 (77.8) ^b^	1 (11.1)	1 (11.1)
CC (N=33)	19 (57.6) ^b^	11 (33.3) ^d^	3 (9.1)

*P* < 0.001, ^a^ compared with ^b^, and ^c^ compared with ^d^.

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
