# Peer review of "Associations among Serum Lipocalin-2 Concentration, Human Papilloma Virus, and Clinical Stage of Cervical Cancer"

_medicina, 2019, doi:10.3390/medicina55060229_

Round 1
Reviewer 1 Report
In this manuscript, the authors explore the serum concentration of LCN2 in cervical cancer and compare to healthy women in a case-control study.
Introduction: clearly lays out the existing data among other tumor sites and identifies the gap in knowledge in cervical cancer
Methods: How did authors identify which HPV types were high-risk and which were not? There are variable strains used in the literature, some citing 14, 15, etc. Recommend a reference to support the use of 15. How was sample size determined?
Results:
There are now new cervical cancer staging guidelines, which ones were used for this manuscript? The older FIGO guidelines only include clinical staging which may skew the results when comparing early vs later stages. For example, using only clinical stage criteria, a patient may have Stage IBI SCC with a positive lymph node which has a worse prognosis than a patient with Stage IBI SCC without positive lymph nodes. It is possible that the LCN2 may be greater in the node positive population, despite the lower clinical stage. Also, is it possible to have information in histology of cervical cancer? It would be interesting to determine if squamous cell caricinoma vs adenocarcinoma etc have differences in LCN2 and its ability to predict. Figures well done and easy to read.
Author Response
How did authors identify which HPV types were high-risk and which were not? There are variable strains used in the literature, some citing 14, 15, etc. Recommend a reference to support the use of 15. How was sample size determined?
Thank You for the review and comments.
For HR HPV types identification we used standardised kit (DiaMex, USA) for in vitro diagnostic use (IVD) consuming in routine practice as well. Also we used literature to justify this high - risk HPV types, for example:
Muñoz N, Bosch FX, de Sanjosé S, Herrero R, Castellsagué X, Shah KV, Snijders PJ, Meijer CJ; International Agency for Research on Cancer Multicenter Cervical Cancer Study Group. Epidemiologic classification of human papillomavirus types associated with cervical cancer. N Engl J Med. 2003 Feb 6;348(6):518-27.
If You have in mind size of the tissue it was 3 - 5 mm tissue fragment from the most affected cervix place also from the same place histological approvement was received (HPV was found in all cases of CC).
There are now new cervical cancer staging guidelines, which ones were used for this manuscript? The older FIGO guidelines only include clinical staging which may skew the results when comparing early vs later stages. For example, using only clinical stage criteria, a patient may have Stage IBI SCC with a positive lymph node which has a worse prognosis than a patient with Stage IBI SCC without positive lymph nodes. It is possible that the LCN2 may be greater in the node positive population, despite the lower clinical stage. Also, is it possible to have information in histology of cervical cancer? It would be interesting to determine if squamous cell caricinoma vs adenocarcinoma etc have differences in LCN2 and its ability to predict. Figures well done and easy to read.
In spite of the fact that our study was performed during 2017 - 2018 years for data analysis revised FIGO staging classification of carcinoma of the cervix uteri was used (2018). We added this information to the manuscript (section Matherials and methods). All cases with positive lymph nodes detected by imaging techniques (MRI, CT or ultrasound) or pathological findings were allocated to stage III. Due to small amount of cases we didn’t divide results in substages IIIA, IIIB, IIIC1, IIIC2. We used for comparison only st. I, II, III, IV.
Talking about histology only cases with squamous cells carcinoma were analysed (we added this information to the article).
Reviewer 2 Report
Lipocalin 2 (Lcn2) has gained recent attention as both a potential biomarker and a modulator of cervical cancers (CC) which is caused by sexually acquired infection with human papilloma viruses (HPV). The aim of the study was to determine whether serum LCN2 could serve as diagnostic marker of CC and to evaluate the correlation between its serum concentration, the clinical stage of the cancer and HPV infection in women.” Authors found that serum LCN2 could be associated with advanced stage of CC and with HPV infection irrespective of the CC.
Major:
The study is based on the analysis of serum, liquid-based cytology medium (LBCM) and tissue biopsy samples collected from 33 patients with histologically proven cervical cancer (CC), 9 patients with high-grade cervical intraepithelial neoplasia (HSIL) and 48 control patients negative for intraepithelial lesions or malignancy (NILM). The sample size is too small to differentiate between low and high grade CCs, but perhaps sufficient to do the analysis of 33+9 CC and 48 control patients with and without HPV infections. Therefore, one of the conclusions “ LCN2 can be used to identify patients with a poor prognosis who may benefit from more aggressive management” should be omitted. The same applies to HPV types.
Minor:
1. Please proof-read the text. For example correct the sentences in the abstract in the abstract the following way: “The aim of our study was to determine whether serum LCN2 could serve as diagnostic marker of cervical cancer (CC) and to evaluate the correlation between its serum concentration, the clinical stage of the cancer and HPV infection in women”, “An 18 HPV genotyping kit was used for detection and differentiation of 15 high-risk (HR) HPV types in the liquid-based cytology medium (LBCM) and tissue biopsy. “Several types of HR HPV were found in the LBCM compared to tissue biopsy (P=0.044).” And the following sentence: “ HPV16 was prevalent in tissue biopsy compared to LBCM (P<0.05).
2. What is shown on axes X of Fig. 1 and 2? Please add axes names.
3. Please omit repetitions such as “shows that many HR HPV types together with HPV16/18 can colonize the vagina and the cervix, but often HPV16 alone penetrates in the tissue and causes CC” in discussion and “show that many HR HPV types together with HPV16/18 can colonize the vagina and the cervix, but often HPV16 alone penetrates in the tissue and causes CC” in conclusion.
4. Please use the same font and size for all the figs.
Author Response
The study is based on the analysis of serum, liquid-based cytology medium (LBCM) and tissue biopsy samples collected from 33 patients with histologically proven cervical cancer (CC), 9 patients with high-grade cervical intraepithelial neoplasia (HSIL) and 48 control patients negative for intraepithelial lesions or malignancy (NILM). The sample size is too small to differentiate between low and high grade CCs, but perhaps sufficient to do the analysis of 33+9 CC and 48 control patients with and without HPV infections. Therefore, one of the conclusions “LCN2 can be used to identify patients with a poor prognosis who may benefit from more aggressive management” should be omitted. The same applies to HPV types.
Thank You for the review and comments.
We aren’t sure if it be correct to analyse invasive disease cases (n=33) together with benign condition (n=9) - intraepithelial neoplasia, not invasive disease. We aware about unequal groups, but we used random function to get equal control group and the results were the same as described in the article. Also we calculated the power of the study and it was 0,8 proving that the analysis can be made.
We agree that our conclusions were too strong. We corrected them:
LCN2 probably my be used to identify patients with more advanced disease required more aggressive treatment.
Minor:
1. Please proof-read the text. For example correct the sentences in the abstract in the abstract the following way: “The aim of our study was to determine whether serum LCN2 could serve as diagnostic marker of cervical cancer (CC) and to evaluate the correlation between its serum concentration, the clinical stage of the cancer and HPV infection in women”, “An 18 HPV genotyping kit was used for detection and differentiation of 15 high-risk (HR) HPV types in the liquid-based cytology medium (LBCM) and tissue biopsy. “Several types of HR HPV were found in the LBCM compared to tissue biopsy (P=0.044).” And the following sentence: “ HPV16 was prevalent in tissue biopsy compared to LBCM (P<0.05).
Was corrected.
2. What is shown on axes X of Fig. 1 and 2? Please add axes names.
It seems that during the formatting process Figures were distorted. We put our original ones again.
3. Please omit repetitions such as “shows that many HR HPV types together with HPV16/18 can colonize the vagina and the cervix, but often HPV16 alone penetrates in the tissue and causes CC” in discussion and “show that many HR HPV types together with HPV16/18 can colonize the vagina and the cervix, but often HPV16 alone penetrates in the tissue and causes CC” in conclusion.
The repetitions was removed.
4. Please use the same font and size for all the figs.
We checked the figures ones again and put them again.
Round 2
Reviewer 2 Report
Please correct concluding sentence: "LCN2 probably my be used to identify patients with more advanced disease required more aggressive treatment." to "LCN2 could be used to identify patients with advanced disease requiring more aggressive treatment."
Author Response
Sentence was corrected.